# Measurement Invariance and Construct Validity of the Satisfaction With Life Scale (SWLS) in Community Volunteers in Vietnam

**DOI:** 10.3390/ijerph19063460

**Published:** 2022-03-15

**Authors:** Willem A. Arrindell, Irene Checa, Begoña Espejo, I-Hua Chen, Danilo Carrozzino, Phuong Vu-Bich, Huong Dambach, Paula Vagos

**Affiliations:** 1Faculty of Psychology, University of Social Sciences and Humanities, Vietnam National University-HCM, Ho Chi Minh City 700000, Vietnam; phuongvu@hcmussh.edu.vn (P.V.-B.); huong.dambach@hcmussh.edu.vn (H.D.); 2Department of Behavioral Sciences Methodology, University of Valencia, 46010 Valencia, Spain; irene.checa@uv.es (I.C.); begonya.espejo@uv.es (B.E.); 3Chinese Academy of Education Big Data, Qufu Normal University, Qufu 273165, China; ahole.chen@gmail.com; 4Department of Psychology “Renzo Canestrari”, University of Bologna, 40126 Bologna, Italy; danilo.carrozzino@unibo.it; 5Instituto de Desenvolvimento Humano Portucalense, Universidade Portucalense Infante D. Henrique, 4200-072 Porto, Portugal; paulavagos@ua.pt

**Keywords:** Satisfaction With Life Scale, subjective well-being, measurement invariance, confirmatory factor analysis, health status, marital status, income, Vietnamese population

## Abstract

Worldwide, the Satisfaction With Life Scale (SWLS) has become the most widely used measure of life satisfaction. Recently, an authorized Vietnamese-language version has been introduced. Using a convenience sample comprising community volunteers from Ho Chi Minh City (N = 1073), confirmatory support was found for the cross-national constancy of the one-dimensional structure underlying the SWLS. Corrected item–total polyserial correlations and Omega coefficient were satisfactory. Using multi-group confirmatory factor analysis, configural, metric, and scalar invariance of the SWLS factorial structure were tested by gender, age, marital status, income, and educational level. Strong evidence of scalar invariance was found for gender and education, on which relevant subgroups did not differ in terms of latent means. Partial scalar invariance was found for marital status (item 4 and 5) and income (item 4). Being involved in an intimate relationship or having a higher income were associated with higher latent means. Scalar invariance in relation to age was very poor. Accordingly, caution must be exerted when comparing age groups. A high SWLS score was predictive of good self-rated health. Implications of the findings are briefly discussed.

## 1. Introduction

Realizing that measurement issues related to subjective well-being (SWB) were understudied and pivotal, Diener, Emmons, Larsen, and Griffin [1] developed the 5-item Satisfaction With Life Scale (SWLS) to form the foundation for future scientific work. With a citation count exceeding 30,000, the SWLS has been translated into 39 different languages (see eddiener.com/scales, accessed on 1 March 2022) and has become the most widely used measure for assessing the life satisfaction component of SWB [2,3,4]. This component is simply defined as a cognitive, overall evaluation of one’s life [5].

The SWLS is not only widely used by behavioral scientists and public health professionals [2,3] but also by happiness economists [6]. The measure has also received attention from philosophers [7], and it is popular in national [3,8,9] and international [3,10] surveys of well-being. Special versions of the SWLS have been administered to children [11], adolescents [12], adults [3,13], and the elderly [3,9]. The SWLS has even been translated into American Sign Language [14]. The measure exists with different response options [15] or temporal orientation [3] and has not only been administered to healthy (non-patient) participants in (inter)national surveys but also to special populations [3], including psychiatric patients [3,16], breast cancer patients [17], people with traumatic brain, spinal cord, or burn injury [18]; persons with multiple sclerosis [19], and patients with Parkinson’s disease [20]. SWLS scores have been shown to possess satisfactory reliability as estimated with McDonald’s omega [17]. In addition, as predicted, the SWLS has been shown to possess moderate temporal (test-retest) stability [3], to correlate meaningfully with measures of mental health [3,15,17], and to discriminate in predicted direction between samples comprising psychiatric patients and individuals with health concerns on the one hand and healthy controls on the other hand [3,16]. In therapy, the SWLS has been shown to be sufficiently sensitive to detect improvement (higher levels of satisfaction with life) over the course of intervention [3]. Furthermore, lower scores on the SWLS have been demonstrated to longitudinally predict higher risk of psychopathology and self-destructive behavior, for example, suicide 20 years later, even after controlling for other risk factors, such as substance use, age, and gender [3]. Using the SWLS as an outcome measure, it was also found that greater goal instability assessed at the onset of disability of people experiencing serious health concerns was predictive of lower satisfaction with life a year later [3]. It should also be pointed out that Kjell and Diener [21] showed that a short, three-item form of the SWLS demonstrates as strong psychometric properties as the original five-item scale.

The robustness and internal validity of many of the findings reported above depend heavily on measurement invariance, which assesses the (psychometric) equivalence of a construct across groups or measurement occasions and thereby demonstrates that a construct has the same meaning to those groups or across repeated measurements [22]. To exemplify the consequence of measurement noninvariance, Putnick and Bornstein [22] provided the following illustration in their study of depression in women and men. Assume that frequency of crying, weight gain, and feelings of hopelessness are indicative of the severity of depression in women, but only feelings of hopelessness are indicative of the severity of depression in men. If the three indicators are combined into a scale to compare depression in women and men, mean differences on the scale may mislead because crying and weight gain have little relation to depression in men. In this example, men may score lower than women on the relevant scale because they cry less and gain less weight. However, crying and weight gain are not associated with depression in men in the first place. Hence, noninvariance of a construct across groups and measurements can lead to erroneous conclusions, and the same problem would apply in a therapy evaluation study if the intervention, treatment protocol, or trial could change the way participants interpret the measuring constructs under study [22]. Thus, it is important to have information on measurement invariance so that one does not misinterpret results.

In relation to dimensionality of the SWLS, Clench-Aas et al. [23] argued that many studies have supported a unidimensional model with the proof that traditional exploratory factor analysis (EFA) indicated that a single latent factor accounted for a majority of the variance in life satisfaction scores when in fact, confirmatory factor analysis (CFA) should have been the preferred analytic tool in view of the fact that there is a well-founded hypothesis and expectation about dimensionality [1]. Studies in which CFA was used found proof of a one-dimensional model, which clearly suggested that the same weight matrix (scoring key) employed in the country in which the SWLS originated (USA) could be transported to the new national group, for example, to Spain [24], Germany [11], Italy [25], Lithuania [26], and Greece [27]. These results are consistent with previous EFA evidence obtained in other populations [2,3,23]. However, straightforward, cross-cultural comparisons in terms of factor loading patterns or mean scale scores on the SWLS with data from comparable groups from the USA were neither intended nor yet carried out in those studies that employed EFA [2,3,11,23,24,25,26,27].

To test measurement invariance across participants from various groups (nationality, culture, gender, age, etc.), researchers use a statistical technique termed “multigroup confirmatory factor analysis” (MG-CFA [28]). In doing so, three typical phases of factorial invariance are tested: configural invariance (does the stipulated overall factor structure fit well across subgroups in one’s sample?), metric invariance (does each item of a scale load onto the specified latent factor in a similar manner and with similar magnitude across subgroups?), scalar invariance (are item intercepts equivalent across subgroups?). When metric invariance is demonstrated, one can assume that differences in factor variances and covariances are not attributable to subgroup differences in the properties of the scales themselves. Ascertaining scalar invariance allows one to substantiate multi-group comparisons of factor means, and one can be confident that statistically significant differences in group means are not due to differences in scale properties across subgroups (for example, between males and females or at different ages). These steps are necessarily sequential, and researchers typically stop testing when any of these steps produces evidence of noninvariance. Researchers would then examine the factor loadings and item intercepts on an item-by-item basis to determine which items are the main contributors toward measurement noninvariance. More specifically, Putnick and Bornstein [22] noted that failing to find scalar invariance leaves at least one alternative option (rather than assuming that the relevant construct is noninvariant and merely discontinuing invariance and group difference testing), namely to investigate the source of noninvariance by sequentially releasing (in a backwards approach) or adding (in a forward approach) item intercept constraints and retesting the model until a partially invariant model is achieved. Illustrative is the study by Jang et al. [10], who examined measurement invariance of the SWLS in 7004 managers from local companies in 26 countries (39% females) using three methods of analyses, including MG-CFA. The countries were Chile, the Netherlands, Finland, Puerto Rico, Peru, Greece, New Zealand, Turkey, Argentina, Taiwan, Canada, Bolivia, Estonia, Slovenia, Poland, Australia, Romania, the United States, the United Kingdom, Hong Kong, Korea, China, Ukraine, Japan, Bulgaria, and Spain. Jang et al. demonstrated that configural and metric invariance of life satisfaction held across 26 countries, whereas scalar invariance did not. Further analyses showed with partial invariance testing that the intercepts of three items (2, 4, and 5) were noninvariant (see Section 2.2 for the specific SWLS items). Accordingly, based on two invariant intercepts, factor means of countries were compared.

For the period spanning from January 1985 to August 2016, Emerson, Guhn, and Gadermann [29] comprehensively identified, reviewed, and assessed research describing measurement invariance of the SWLS using MG-CFA across various subgroups. They identified 27 articles representing 66,380 respondents across 24 nations. Gender, age, and culture were the most common types of measurement invariance assessed. The highest level of invariance tested in each article varied. Focusing on the three levels discussed above, Emerson et al. [29] found that findings generally supported a unidimensional structure (configural invariance), but less commonly supported were equivalent factor loadings (metric invariance). They noted that over half of the gender invariance analyses supported (at least) scalar invariance. However, scalar invariance was supported in only one of the nine studies on invariance in relation to age. The authors concluded that comparisons across gender on the SWLS may be valid in some situations but most likely not across culture or age groups.

For the purpose of the present study, the major focus will be on the background, demographic variables most studied to date, namely gender and age. Partly in agreement with Emerson et al. [29], more recent studies support the measurement invariance characteristics of the SWLS in relation to both age and gender. For example, at least six studies conducted in Spanish speaking countries have found evidence of measurement invariance in relation to gender and age. Esnaola et al. [30] found support for at least scalar invariance for both age and gender in a mixed sample of 701 Spanish and Mexican adolescents (52.9% females); so did Checa, Perales, and Espejo [31] in an adult sample comprising students and community subjects (55.2% females) from Spain and also Ruiz et al. [32] in a mixed (combined) Colombian sample of undergraduates, students, and participants from the general population and subjects undergoing treatment for an emotional disorder (62%, 74%, and 52% females, respectively). Lorenzo-Seva et al. [33] also found evidence for such in Spain in a sample of 713 adult oncology patients (57.8% females), as did Espejo et al. [15] in 1255 adult participants from Colombia (64.2% females) and Ortuño-Sierra et al. [34] in 1020 non-clinical adolescent Spanish participants (61.6% females).

Moreover, a study by Arikan and Zorbaz [35] carried out in Turkey among 500 university students (73.2% females) also demonstrated at least scalar invariance in relation to gender. Garcia et al. [36] also in a Swedish study with 264 psychiatric patients spanning a wide age range from late adolescence to the elderly (approximately 47% females) found evidence of invariance with respect to gender. In addition, using the Serbian version of the SWLS with a mixed sample of 2595 secondary school adolescents, undergraduate students, and community adults (62.2% females), Jovanović [37] found evidence of full scalar invariance across gender and partial scalar invariance across age. In a further study with the SWLS in adolescent samples conducted across 24 countries and regions (22,710 participants, 53% females), Jovanović et al. [38] found support for full scalar invariance of the SWLS across gender in the pooled sample. This indicates that the corresponding items operated similarly among girls and boys in the full sample. However, further analyses within each group revealed complex findings, suggesting that invariance of the SWLS across gender depends on culture. More specifically, full scalar invariance was supported in 10 (Argentina, Bulgaria, Finland, Hong Kong, Japan, Malaysia, Russia, South Africa, Taiwan, and Turkey) out of 24 countries and regions, full metric invariance in 8 countries (China, Italy, India, Romania, Serbia, South Korea, Spain, and the United Kingdom), and configural invariance was found in 3 countries (Indonesia, Portugal, and Switzerland). Jovanović et al. concluded that this suggests that gender invariance of the SWLS among adolescents should not be taken for granted and that testing for gender invariance of the SWLS is a necessary step prior to examining gender differences in mean levels and correlates of life satisfaction measured with the SWLS. Their evaluation of measurement invariance across age largely supported invariance of the SWLS during adolescence.

Fewer studies have addressed measurement invariance in relation to marital status, education, and income.

With Spanish adults, Checa et al. [31] found evidence of scalar invariance and Bai et al. [8] in their representative Chinese sample of partial scalar invariance in relation to educational level. Bai et al. [8] also found evidence of partial scalar invariance across income. In addition, Checa et al. [31] also found evidence of partial scalar invariance in relation to marital status.

### The Present Study

Recently, an authorized Vietnamese-language version of the SWLS has become available. Even though a handful of studies have published internal consistency reliability figures based on Cronbach’s alpha for a Vietnamese translation of the SWLS [39,40,41,42], no attempts have as yet ever been made to thoroughly psychometrically evaluate a Vietnamese translation of the SWLS. Cronbach’s alpha is neither an optimal measure of reliability [43], nor does it provide an adequate indication of a measure’s one-dimensionality [44]. Accordingly, the first aim of the present study was to determine through CFA whether the authorized Vietnamese SWLS represents a measure of a single construct as advanced by its constructors in the USA [1]. The second aim of the present study was to address the practical limits (if any) of generalizability of factor structure and mean scores across important participant parameters, including gender, age, marital status, income, and educational level. This entailed the test of measurement invariance of the SWLS in relation to these five parameters.

Provided that evidence was obtained for at least partial scalar invariance, the third aim of the present study was to determine background, demographic correlates of the Vietnamese-language SWLS by calculating its associations with gender, age, income level, educational level, and marital status. The following predictions in relation to each background factor were tested.

Studies with the SWLS have shown that the general pattern has been for happiness to be similarly available to people of any gender and age [45]. The association between SWLS and income is complex: it may not exist [45], it may be linear up to a certain threshold as demonstrated in data collected in 26 European countries [46], or it may rise linearly [9], with an equally steep slope even above a substantially high income (USD 80,000 per year) as below it, as was recently demonstrated by Killingsworth [47]. As educational level may correlate with income level [48], it was further hypothesized that a higher education level would coincide with higher SWLS scores. However, Diener et al. [49] reported that much of the usual association (effect size) of around 0.1–0.2 with education may be due to the correlation of education with occupational status (not measured in the present study) and income (see Discussion).

It was further hypothesized that a positive relationship would be obtained between SWLS and self-perceived good self-rated health. These associations in relatively large samples are generally small but statistically significant, for example, as low as +0.14 with “not suffering from long-term illnesses” and +0.3 with “self-reported health” in the Statistics Netherlands’ 2010 Perceptions Survey in over 3000 Dutch participants [50] or +0.2 with “general health” in the Mexican Health and Aging Study in 13,220 Mexican adults [51,52].

A positive relationship would be expected between SWLS and the state of being married or involved in an intimate relationship. In a sample of 59,169 persons in 42 nations, Diener et al. [53], for example, found that the relations between marital status and subjective well-being were very similar across the world; although cultural variables were found to alter the size of the relevant relationships, the effect sizes were very small.

Overall support for the relevant correlates would further strengthen evidence in favor of construct validity of the Vietnamese-language SWLS.

## 2. Materials and Methods

### 2.1. Subjects and Procedure

Participants were recruited in Ho Chi Minh City (HCMC), District 1, in both the public (squares, parks, public libraries) and social space (social center or other gathering place where people gather or interact, including community pubs, restaurants, cafes, shopping malls). Between September 2016 and April 2017, individuals with Vietnamese nationality were invited to participate in a study about their personal life. They were asked to fill out a brief form anonymously and individually and were informed that their data would be treated confidentially and only be used for scientific purposes. This was done through an information sheet, and participants were given the opportunity to ask questions about the background of the study. There were no legal, social, political, economic, or health risks to the subjects involved in the study. By filling out the questionnaire set, all participants agreed to participate in the study (combined verbal and implied consent). In spite of the fact that participation in the study by a minority of minors (early secondary school pupils) was dependent on approval by the parent(s), after which a questionnaire could be voluntarily returned, refusal occurred in only a few cases. The sample should be considered one of convenience. The sample comprised 1073 individuals aged 14 to 64 years (M = 23.44; SD = 6.52).

Demographic data on gender, marital status, educational level, and income are depicted in Table 1.

### 2.2. Measures and Translation

Satisfaction With Life Scale (SWLS; [1]). For measuring overall life satisfaction, the SWLS contains five positively phrased items: (1) “In most ways, my life is close to my ideal”; (2) “The conditions of my life are excellent”; (3) “I am satisfied with my life”; (4) “So far, I have gotten the important things I want in life”; and (5) “If I could live my life over, I would change almost nothing”. Each item is rated on a 7-point-rating scale ranging from 1 (“strongly disagree”) to 7 (“strongly agree”).

Self-rated health refers to a one-item question that was formulated as “in general, would you say that your health is ‘excellent’, ‘very good’, ‘good’, ‘fair’, or ‘poor’?” It captures people’s own assessment of health according to their own definition of health.

Both measures were translated from English to Vietnamese by two independent, bilingual Vietnamese translators and back-translated into English by a third bilingual sworn translator by the method advised by Brislin, Lonner, and Thorndike [54]. In case of any difference, an independent fourth expert (bilingual psychologist) was consulted to ensure that the meaning of each item was correctly captured and to finally arrive at one consensus translation.

### 2.3. Statistical Analyses

A Confirmatory Factor Analysis (CFA) was performed for testing the one-factor model underlying the items of the SWLS. For assessing measurement invariance across various subgroups, multi-group confirmatory factor analysis (MG-CFA) was employed. Chi-square (χ^2^), the comparative fit index (CFI), the root-mean-square error of approximation (RMSEA), and the standardized root-mean-square residual (SRMR) were used to evaluate goodness of fit. Cut-offs for goodness of fit were 0.90 for CFI and 0.08 for RMSEA/SRMR, indicating acceptable fit, and 0.95 for CFI and 0.06 for RMSEA/SRMR, indicating excellent fit [55].

A structural equation model was specified to study the construct validity of the Vietnamese SWLS by using the general health rating as covariate of the latent SWLS variable, specified by its five items.

Factorial invariance was evaluated by gender, age, marital status, educational level, and income. Three nested models with increasing degree of restriction were tested: the base model assessed configural invariance and allowed free estimation of all the parameters for each group. The metric (weak) invariance model, nested in the configural model, added the restriction of invariant factor loadings among groups. Finally, the scalar (strong) invariance model, nested in the second model, added the intercept constraint of the invariant items among the comparison groups.

For testing loading invariance, a change of ≥0.010 in CFI supplemented by a change of ≥0.015 in RMSEA or a change of ≥0.030 in SRMR would indicate that there is no invariance; for testing intercept invariance, a change of ≥0.010 in CFI supplemented by a change of ≥0.015 in RMSEA or a change of ≥0.010 in SRMR would indicate that there is no invariance [56].

Factor measurement reliability was evaluated with the ω coefficient [57]. In general, an omega value ≥0.70 is considered acceptable [58]. In addition, to estimate homogeneity of SWLS items, the mean inter-item correlation and corrected item–total polyserial correlations were calculated [59].

CFA, measurement invariance, construct validity, and corrected item–total polyserial correlations were determined with Mplus 8.6 [60]. Maximum Likelihood Robust (MLR) was used to estimate model parameters and statistics of the model, as it is considered the best estimator with missing data and with Likert scales of more than 5 points [61].

For determining descriptive statistics for the socio-demographic variables, for the SWLS items, and total score, IBM SPSS 26 was used.

Cohen’s measure of effect size was estimated in line with Cohen [62]. This measure of effect size is an index of the magnitude or importance of an association (r) or between-group difference (d). Unlike a *p*-value, an effect size can be used to quantitatively compare the results of studies conducted in different labs by different researchers at different times. For purposes of interpretation, Cohen [62] considers r = 0.10 or d = 0.20 as “small”, r = 0.30 or d = 0.50 as “medium”, and r = 0.50 or d = 0.80 as a reflection of “large” effect size.

## 3. Results

### 3.1. Descriptives and Reliability

Table 2 shows descriptive statistics (means and standard deviations) for each item, for the overall SWLS, and for the total sample. Skewness values were between −0.06 and 0.24 for the items and –0.36 for the total score. Regarding kurtosis, the values were between −0.12 and −1.37 for the items and −0.37 for the total score.

The SWLS items were all significantly inter-correlated, 0.30–0.58 (*p* < 0.01), with a mean inter-item correlation (homogeneity) of 0.38. Item–total polyserial correlations were satisfactory and ranged from 0.43 to 0.64 (standard errors 0.014–0.021).

### 3.2. CFA

The one-dimensional model for the total sample presented an adequate fit, with χ^2^ (5) = 33.721, CFI = 0.968, RMSEA = 0.073, RMSEA 90% confidence interval = (0.051, 0.097), SRMR = 0.027. All factor loadings were statistically significant (*p* < 0.01) (see Figure 1).

The ω index value for the SWLS was satisfactory, 0.756.

### 3.3. Association with Perceived Health

When studying the association between the SWLS and perceived health, structural equation modeling presented an adequate fit, with χ^2^ (9) = 44.454, CFI = 0.964, RMSEA = 0.061, RMSEA 90% confidence interval = (0.044, 0.079), SRMR = 0.028 (see Figure 2). In this model, all factor loadings for the SWLS were statistically significant (*p* < 0.01) as well as the correlation between the latent variable SWL and the general health item (r = 0.184, *p* < 0.01).

### 3.4. Measurement Invariance

Marital status, age, income, and educational level were dichotomized. The age variable was arbitrarily divided into two groups: 14–24 y and 25–35 y. The rest of the sample (up to 64 years) accounted for less than 5% of the total; therefore, this subgroup was not taken into account. These two subgroups, mainly comprising adolescents and young adults, were considered in view of their potentially overlapping developmental characteristics as opposed to a relative non-overlap of such characteristics with the remaining small adult subgroup. The determine the two income subgroups, a cut-off score of 6 million Vietnamese Dong (VND) per month was employed. This cut-off was set at VND 6 million per month because the average monthly urban salary in the study period 2016-2017 was around VND 5.4 million (https://www.ceicdata.com/en/vietnam/average-monthly-earning-quarterly/average-monthly-earning-urban, accessed on 7 June 2021). It should be pointed out that in that same time period, VND 1 = USD 0.000044, or VND 1 million = USD 44 (approximately). Accordingly, the low-income subgroup comprised participants with a monthly income up to VND 6 million, and the high-income subgroup comprised participants with a monthly income ranging from VND 6 million up to over VND 10 million. The two marital status subgroups were participants with an intimate partner (married or cohabiting/living with an intimate partner) versus participants without an intimate partner (widowed, divorced, separated, or single). Two education subgroups were established: participants with a higher education degree (bachelor diploma or higher) versus participants without a higher education degree (no schooling, primary school, professional mid-school, high school, undergraduate bachelor). Descriptive data for the SWLS for each of the subgroups are also shown in Table 2.

As a prerequisite for the measurement invariance test, in Table 3 are shown the goodness-of-fit indices for the one-dimensional model in relation to gender, age, marital status, educational level, and income. In all subgroups, the model showed a good fit. The nested invariance models are presented below in ascending order of level of restriction.

The SWLS showed strong invariance by gender and by educational level. This means that the corresponding latent means can be compared. When the latent mean values were set to zero for men, there were no differences in life satisfaction by gender (b = 0.069, z = 1.523, *p* = 0.128). The latent mean values were set to zero for participants in group with no university degree. There were no differences in life satisfaction by educational level (b = 0.029, z = 0.612, *p* = 0.540).

The SWLS did not show scalar invariance by age. After analyzing the differences between intercepts of the metric and scalar models, a relevant difference was observed in relation to items 2 and 4. Accordingly, a partial scalar model was tested, freeing the constraint of items 2 and 4 since these were the items that presented the greatest difference between the two age groups (see Table 3). Still, even though the RMSEA and SRMR criteria were met (change of <0.015 in RMSEA and change of <0.010 in SRMR), the CFI decreased by 0.017. Therefore, it was not possible to compare the relevant latent means.

In relation to marital status, the SWLS did not show scalar invariance. After analyzing the differences between intercepts of the metric and scalar models, a relevant difference was observed in relation to items 4 and 5. Thus, a partial scalar model was tested, freeing the constraint of items 4 and 5 since these were the items that presented the greatest difference between participants with and participants without an intimate partner (see Table 3). In doing so, invariance criteria were met, and the latent means could be compared. The latent mean values were set to zero for the group with an intimate partner. A one-tailed test showed differences in life satisfaction by marital status (b = −0.107, z = −1.743, *p* < 0.05; Cohen’s d = 0.16 or practically small effect size) such that participants with an intimate partner reported greater satisfaction with life than those without an intimate partner.

Scalar invariance did not present an acceptable fit by income. Following the analysis of differences between intercepts of the metric and scalar models, a meaningful difference was observed in relation to item 4, so a partial scalar model was tested releasing the constraint of element 4, as this item evidenced the greatest difference between the low- and high-income group (see Table 3). The latent mean values were set to zero for the low-income group. Subjects with relatively higher incomes showed better life satisfaction than their counterparts with relatively lower incomes (b = 0.448, z = 5.543, *p* < 0.001; Cohen’s d = 0.20 or small effect size).

Further analyses revealed that the lack of parallel between the education and income findings could be due to an age difference in the magnitude of the relationship between the two first-mentioned variables; polychoric correlations between education and income increased with increasing age, from practically small via medium to large effect size as follows: 14–24 y (N = 723, r = 0.06, CI: 0.11, 0.03), 25–35 y (N = 300, r = 0.31, CI: 0.40, 0.21), and 36–64 y (N = 50, r = 0.65, CI: 0.86, 0.43).

## 4. Discussion

In line with previous findings [23], CFA for the total sample showed that the original U.S. scale composition for the SWLS can be transported to the Vietnamese culture and that the same weight matrix (scoring key) may in principle be employed with Vietnamese community subjects. Further in line with previous studies [17,34], the Vietnamese-language SWLS was shown to reliably measure what it intends to measure when McDonald’s omega was used as the estimate for reliability.

Replicating findings from a series of studies on the measurement invariance of the SWLS in relation to gender [29,30,31,32,33,34,35,36,37,38], clear evidence was found for scalar invariance with, as predicted [45], similar average life satisfaction scores for females and males.

Using a nationally representative sample of China, Bai et al. [8] found evidence of partial strict [22] invariance across education (items 1, 2, and 3), whereas in the present study, scalar invariance was observed. When means of the relevant subgroups were compared, unlike what was predicted, highly educated participants were on average not more satisfied with their lives than their less-educated counterparts. This observation did not run parallel to the findings in relation to income, where having a higher income was associated with higher SWLS scores [9] probably because of the complex combined influences: (a) of age on SWB itself [9,54], (b) of age on the relationship between education and income observed in the present study, and (c) of other potential factors that were not measured in the present study. Following Diener, et al. [49], such unmeasured factors would include employment status and work aspirations. Much of the relationship between education and SWB, in their analysis of the evidence, was due to the correlation of education with occupational status and income; education may be only indirectly related to SWB ([49], p. 293). Importantly, Joshanloo and Jovanović [63] demonstrated in their sample of 952,739 adults across 150 countries that predictors of life satisfaction were generally more effective in predicting life satisfaction in older groups (i.e., 34–43, 44–57, and 58 y and older) than in younger groups (i.e., 15–24, 25–33 y). It is clear that in the present study, the older groups were under-represented, which could have affected the education–SWLS relationship. Only a repetition and extension of the present study will enable us to be certain on the nature of the relevant relationship.

Additionally, in agreement with predictions (in terms of statistical significance and of effect size), Vietnamese participants involved in an intimate relationship were happier than their equivalents who were single or previously married (i.e., divorced, separated, or widowed). This is the usual finding [53]. Commitment may fulfill basic and universal human needs, provide companionship and freedom from loneliness, lessen the strains encountered in life, increase one’s ability to cope with such strains (sense of mastery), and/or enhance one’s self-esteem and sense of identity [53,64,65]. The present outcome in relation to marital status is not only in tune with Spanish community findings [31] but also in keeping with the findings of a recent study carried out in Vietnam at the Vietnam National Heart Institute among 109 hospitalized adults with congenital heart disease, whose lower scores on the SWLS (life dissatisfaction) were associated with unmarried status [20]. Nevertheless, since there was too wide a difference in subsample size between the relevant subgroups in the present study (approximately 80% versus 20%), this marital status–SWLS outcome should be interpreted with caution; the reason for this being that when measurement invariance is studied, and sample size cells are unbalanced, tests could produce results that are biased because the fit function in the factorial analysis includes a weighting based on the size of each subgroup [66].

Unlike findings from previous studies [12,31,37], in the present study, scalar invariance in relation to age was very poor. Accordingly, an SWLS score of 10 for individuals in the 14–24 y age range does not have the same meaning as a score of 10 for individuals falling in the 25–35 y age range.

In line with expectations, higher scores on the Vietnamese-language SWLS were demonstrated cross-sectionally to have a small effect-sized associations with better perceived health status [9,52] and with higher income [49].

Taken together, the correlates of SWLS described above, in magnitude and sign, support the construct validity of the Vietnamese-language SWLS.

A limitation of the present study was that it did not reflect a representative sample from the Vietnamese society as a whole. Data were collected in only one city of the country (HCMC) and in only one region within HCMC (District 1). In addition, the sample was biased in terms of age in the sense that participants in middle (45–60 y of age) and late adulthood (>60 y) were under-represented. In addition, highly educated participants were over-represented. It should be pointed out that HCMC is mainly urban and densely populated (it takes just up to 0.6% of the country’s land area but contains just over 8% of the population of Vietnam). Moreover, in terms of gross domestic product (GDP) per capita, HCMC is richer than the country’s average level of income; HCMC is the economic center of Vietnam and accounts for a large proportion of the economy of Vietnam. A nationally representative study sample of Vietnam could have yielded entirely different results. As an illustration of the importance of sampling in relation to SWLS measurement invariance, Bai et al. [8], with their nationally representative Chinese sample comprising 4795 participants, found partial scalar invariance across income (items 1, 2, and 3) and residential region with three categories, namely “metropolitan”, “county town”, and “rural area” (items 1 and 2). Bai et al. ([8], p. 195) inferred that people who differ in their social and economic backgrounds in China have different interpretations of the items of the SWLS, especially for the items correlating satisfaction with past life.

Another limitation of the present study has to do with the fact that a classical psychometric approach was used to evaluate the measurement properties of the SWLS. Future studies are needed to assess its transferability, applicability, and dimensionality or scalability using item response theory (IRT) models (i.e., Rasch and Mokken analyses) according to clinimetric patient-reported outcome measures (CLIPROM) criteria [67]. Future studies are also needed to further evaluate the construct validity of the SWLS by examining its relationship with euthymia, an emerging concept in the field of assessment of a subjective state of well-being, which appears to be characterized not only by the absence of affective disorders but, most importantly, also by the presence of positive affect, psychological flexibility, resistance to stress, and a unifying outlook on life, which guides actions and feelings to shape the future accordingly [68,69,70]. Further studies with the SWLS may also benefit from the use of Local Structural Equation Modeling when studying variations in factor model parameters as a function of continuous variables, such as age, education, and income [71].

With the present encouraging findings, researchers can now move ahead with greater confidence with further studies with the SWLS in Vietnam.

## 5. Conclusions

The Vietnamese version of the SWLS, like its original U.S. equivalent, represents a reliable measure of a single construct, which is also predictably associated with perceived general health.

There is evidence of scalar invariance of the SWLS in relation to gender and education and partial scalar invariance in relation to marital status and income but poor evidence of scalar invariance in relation to age. Thus, caution must be exerted when comparing age subgroups on the SWLS.

There is at present sufficient evidence for carrying out further validity and other empirical studies with confidence with the SWLS in the Vietnamese context.

## Figures and Tables

**Figure 1 ijerph-19-03460-f001:**
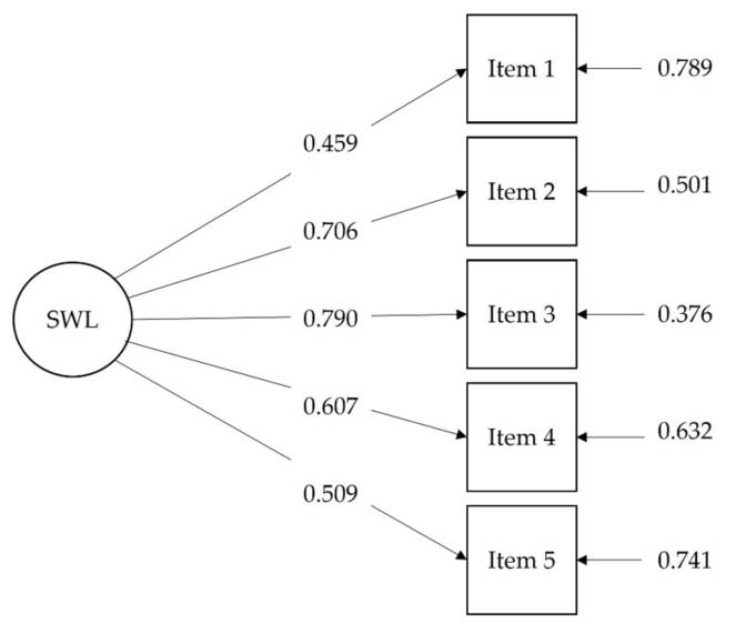
Path model for the Confirmatory Factor Analysis of the Satisfaction with Life Scale (SWLS), with factor loadings and residuals. Standardized coefficients are shown.

**Figure 2 ijerph-19-03460-f002:**
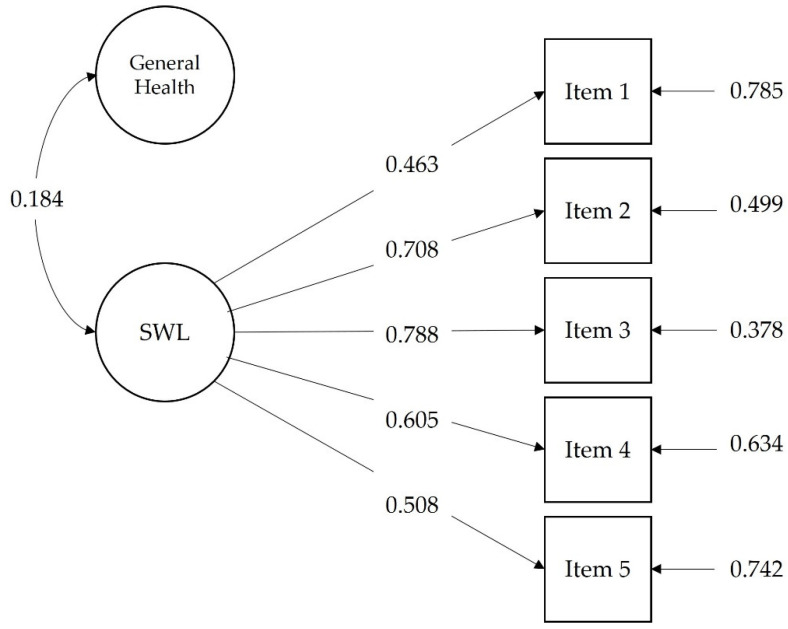
Path model for the association of the Satisfaction with Life Scale (SWLS) with the General Health item. Standardized coefficients are shown.

**Table 1 ijerph-19-03460-t001:** Sociodemographic characteristics of the sample and of the dichotomized groups used to assess measurement invariance.

		*n*	%	Dichotomized	*n*	%
Age				14–24 y	723	67.4%
				25–35 y	300	28%
Gender *	Male	442	41.2%			
	Female	623	58.1%			
Marital status	Married	144	13.4%	With an intimate partner	184	17.1%
	Live with a partner	40	3.7%	Without an intimate partner	889	82.9%
	Widowed	4	0.4%			
	Divorced	16	1.5%			
	Separated	13	1.2%			
	Single	856	79.8%			
Educational level	No schooling	8	0.7%	No university degree	433	40.4%
	Primary School	8	0.7%	University degree	640	59.6%
	Professional Mid-School	25	2.3%			
	High School	246	22.9%			
	Undergraduate Bachelor	146	13.6%			
	Bachelor’s Diploma	69	6.4%			
	Master’s Degree	571	53.2%			
Income (Vietnamese Dong, VND, per month)	No income	328	30.6%			
	Up to 500.000	33	3.1%	Low	443	41.3%
	500K–1 million	30	2.8%	High	302	28.1%
	1–2 million	67	6.2%			
	2–3 million	63	5.9%			
	3–4 million	73	6.8%			
	4–5 million	94	8.8%			
	5–6 million	83	7.7%			
	6–7 million	55	5.1%			
	7–8 million	42	3.9%			
	8–9 million	22	2.1%			
	9–10 million	40	3.7%			
	over 10 million	143	13.3%			

Note: * 8 missing values. Low income, up to VND 5–6 million; high income, VND 6 million and higher. No university degree, no schooling, primary school, professional mid-school, high school, undergraduate Bachelor; University degree, Bachelor’s diploma, Master’s degree.

**Table 2 ijerph-19-03460-t002:** Total score of the Satisfaction with Life Scale (SWLS) and items’ means and standard deviations for the total sample and for the dichotomized groups.

		Item 1		Item 2		Item 3		Item 4		Item 5		SWLS Total Score	
		Mean	SD	Mean	SD	Mean	SD	Mean	SD	Mean	SD	Mean	SD
>Total Sample		4.51	1.43	4.78	1.54	4.96	1.52	3.95	1.63	3.74	1.91	21.94	5.72
Gender	Male	4.57	1.49	4.65	1.55	4.92	1.53	3.87	1.67	3.61	1.99	21.62	5.82
	Female	4.47	1.39	4.87	1.53	4.99	1.51	4.00	1.60	3.82	1.85	22.15	5.63
Age	14–24 y	4.44	1.42	4.88	1.53	4.98	1.52	3.78	1.61	3.62	1.91	21.69	5.54
	25–35 y	4.59	1.43	4.51	1.54	4.92	1.54	4.22	1.62	3.92	1.88	22.16	5.94
Marital Status	With an intimate partner	4.70	1.38	4.88	1.42	5.13	1.43	4.44	1.59	4.23	1.82	23.37	5.59
	Without an intimate partner	4.47	1.44	4.76	1.56	4.93	1.54	3.85	1.63	3.63	1.92	21.64	5.70
Educational level	No university degree	4.37	1.51	4.73	1.66	4.99	1.60	3.88	1.70	3.75	1.94	21.72	6.12
	Bachelor degree or higher	4.60	1.37	4.82	1.46	4.94	1.47	4.00	1.60	3.72	1.88	22.09	5.42
Income	Low	4.38	1.43	4.62	1.56	4.78	1.58	3.78	1.65	3.67	1.90	21.23	5.87
	High	4.71	1.43	4.74	1.47	5.03	1.42	4.33	1.61	3.86	1.88	22.68	5.68

Note: SD, standard deviation; Item 1, In most ways, my life is close to my ideal; Item 2, The conditions of my life are excellent; Item 3, I am satisfied with my life; Item 4, So far, I have gotten the important things I want in life; Item 5, If I could live my life over, I would change almost nothing.

**Table 3 ijerph-19-03460-t003:** Measurement invariance models of the Satisfaction with Life Scale (SWLS).

Models in Each Group		χ^2^	df	Δχ^2^	Δdf	CFI	RMSEA	SRMR	ΔCFI	ΔRMSEA	ΔSRMR
Gender	Male ^a^	17.582 *	5			0.964	0.075	0.031			
	Female	26.681 *	5			0.962	0.083	0.029			
Age	14–24 year ^a^	20.266 *	5			0.971	0.065	0.026			
	25–35 year	8.691 *	5			0.988	0.050	0.022			
Marital status	With an intimate partner ^a^	6.761	5			0.991	0.044	0.027			
	Without an intimate partner	27.385 *	5			0.968	0.071	0.027			
Educational level	Not in Higher Education ^a^	23.311 *	5			0.959	0.085	0.029			
	In Higher Education	15.439 *	5			0.977	0.060	0.029			
Income	Low ^a^	13.296 *	5			0.979	0.061	0.024			
	High	9.609 *	5			0.983	0.055	0.030			
Global models											
Gender											
Configural		51.085 *	10	-	-	0.966	0.088	0.030	-	-	
Metric		53.393 *	14	2.308 (*p* = 0.67)	4	0.967	0.073	0.034	0.001	−0.015	0.004
Scalar		62.833 *	18	9.440 (*p* = 0.05)	4	0.963	0.068	0.037	−0.004	0.005	0.003
Age											
Configural		34.133 *	10	-	-	0.979	0.069	0.025	-	-	
Metric		39.943 *	14	5.810 (*p* = 0.21)	4	0.977	0.060	0.035	−0.002	−0.009	0.010
Scalar		105.310 *	18	65.366 (*p* < 0.01)	4	0.923	0.097	0.049	**0.055**	**0.037**	0.014
Partial scalar		50.305 *	16	10.362 (*p* < 0.01)	2	0.960	0.065	0.043	**0.017**	0.005	0.008
Marital status											
Configural		40.284 *	10	-	-	0.975	0.075	0.027	-	-	
Metric		48.107 *	14	7.823 (*p* = 0.09)	4	0.972	0.067	0.041	−0.003	−0.008	0.014
Scalar		74.106 *	18	25.999 (*p* < 0.01)	4	0.953	0.076	0.049	**−0.019**	0.009	0.009
Partial scalar		50.202 *	16	2.095 (*p* = 0.35)	2	0.971	0.063	0.039	−0.001	−0.004	−0.002
Educational level											
Configural		42.400 *	10	-	-	0.970	0.080	0.029	-	-	
Metric		45.285 *	14	2.097 (*p* = 0.71)	4	0.972	0.067	0.032	0.002	−0.013	0.003
Scalar		57.029 *	18	7.586 (*p* = 0.10)	4	0.969	0.062	0.039	−0.003	−0.005	0.007
Income											
Configural		26.764 *	10	-		0.982	0.067	0.027	-	-	-
Metric		30.597 *	14	3.833 (*p* = 0.42)	4	0.982	0.056	0.035	0.000	−0.011	0.008
Scalar		48.081 *	18	17.484 (*p* < 0.01)	4	0.967	0.067	0.038	**−0.015**	0.011	0.003
Partial scalar		36.348 *	17	5.751 (*p* = 0.12)	3	0.979	0.055	0.039	−0.003	−0.001	0.004

Note: χ^2^, chi-square; df, degrees of freedom; Δχ^2^, chi-square increase; Δdf, degrees of freedom increase; CFI, comparative fit index; RMSEA, root-mean-square error of approximation; ΔCFI, CFI change; ΔRMSEA, RMSEA change; ΔSRMR, SRMR change; * *p* < 0.001; ^a^ reference group; in bold are the change values that exceed the invariance recommendations.

## Data Availability

The raw data file on which the present analyses are based is available from the senior author upon reasonable request.

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
