# Peer review of "Measurement Invariance and Construct Validity of the Satisfaction With Life Scale (SWLS) in Community Volunteers in Vietnam"

_ijerph, 2022, doi:10.3390/ijerph19063460_

Round 1
Reviewer 1 Report
Dear Editor IJRPH,
I would like to thank you for the opportunity you have offered me to evaluate the manuscript entitled: "Measurement invariance and concurrent validity of the Satisfaction With Life Scale (SWLS) in community volunteers in Vietnam" for possible publication in your journal. I have carefully read the manuscript and here are some aspects that may contribute to improve the current version.
The authors present the results of psychometric analyses of the Satisfaction with Life Scale (SWLS) conducted in a large sample of the general Vietnamese population. In addition to evidence of uni-dimensionality in the total sample, the authors show the results of factorial invariance analysis in subgroups of age, gender, marital status, educational level, and income.
Overall, the manuscript is well written and the analyses have been rigorously conducted. However, I believe that the manuscript has possibility for improvement considering some aspects that I mention below.
- First, it is desirable to present the income results in dollars. This would help readers unfamiliar with the Vietnamese money to interpret the description and implications of the factorial invariance results in relation to income level.
- The authors mention that the grouping by age was done through an arbitrary cutoff score, but no justification is given for such a decision and/or no discussion is given as to how this decision may influence the results. This aspect can also be considered for income level.
- It is advisable to incorporate normality statistics to the descriptive statistics of the items. This would help the reader to assess the relevance of using the MLR method as an estimation method for the analyses.
- Regarding marital status, there is a wide difference between the groups (80% - 20% approximately). The possible effect of such a difference on the sensitivity of goodness-of-fit indices in factorial invariance studies is well known. This is an aspect that the authors should incorporate into the manuscript.
- It is necessary to present the p-values associated with the ΔX2. This would help the reader to make a better evaluation of the results.
- I believe that the major limitation of the presentation of the results is that the substantive implications of the results with respect to partial scalar invariance are not discussed. I believe that a slightly more extensive substantive interpretation of why these findings.
- The authors begin the discussion by using the term cross-national consistency. This can lead to confusion, as it can be confused with the existence of cross-national invariance. I suggest using the phrase: "the results are consistent with previous evidence obtained in other populations".
- I think it is important to mention to what extent the magnitude of the relationship between SWLS and general health is similar to previous studies. I also think it is necessary to use the term "perception of general health", which is not the same as "general health".
- Finally, I believe that the use of a classical approach is not the only limitation of the study. It is necessary to incorporate some of the aspects mentioned above. Additionally, they mention the Rasch and/or Mokken analyses, but it is not clear why they do not present these results.
In summary, I believe that the manuscript is well written, clear and synthetic, and considers the main aspects that should be incorporated in this kind of work. However, I believe that the manuscript would benefit from incorporating the recommendations presented. That is why my recommendation is: "accepted with minor modifications".
I thank you again for giving me the opportunity to contribute to your journal.
Best regards.
Author Response
- First, it is desirable to present the income results in dollars. This would help readers unfamiliar with the Vietnamese money to interpret the description and implications of the factorial invariance results in relation to income level.
Good suggestion. Thank you. This has been done where the subgoups in terms of low and high income are defined.
- The authors mention that the grouping by age was done through an arbitrary cutoff score, but no justification is given for such a decision and/or no discussion is given as to how this decision may influence the results. This aspect can also be considered for income level.
Thanks for the suggestion. A better explanation of the age groups is added to the description of the cut-off points in the Measurement invariance of Results section .
- It is advisable to incorporate normality statistics to the descriptive statistics of the items. This would help the reader to assess the relevance of using the MLR method as an estimation method for the analyses.
Thanks for the suggestion, an explanation about the skewness and kurtosis of the items and the total score of the complete sample is included in the text.
- Regarding marital status, there is a wide difference between the groups (80% - 20% approximately). The possible effect of such a difference on the sensitivity of goodness-of-fit indices in factorial invariance studies is well known. This is an aspect that the authors should incorporate into the manuscript.
Thank you for this comment. Yes, when the sizes of both groups are very different, there may be problems in the fit indices obtained when studying the measurement of invariance (Yoon et al., 2018), since inappropriate conclusions can be reached. Largely unbalanced sample sizes can lead to incorrect invariance conclusions because the fit function includes weighting based on sample size. In these cases, it may happen that there is no measurement invariance, and the fit indices are not capable of detecting it, especially if it involves two seriously unbalanced groups. This issue has been included in the Discussion section.
Yoon, M., & Lai, M. H. (2018). Testing factorial invariance with unbalanced samples. Structural Equation Modeling: A Multidisciplinary Journal, 25(2), 201-213. doi: 10.1080/10705511.2017.1387859
- It is necessary to present the p-values associated with the ΔX2. This would help the reader to make a better evaluation of the results.
Thanks for the suggestion. Added reference to p-values associated with the x2 in Table 3.
- I believe that the major limitation of the presentation of the results is that the substantive implications of the results with respect to partial scalar invariance are not discussed. I believe that a slightly more extensive substantive interpretation of why these findings.
Thanks for the suggestion. This issue has been commented on in the discussion section .
- The authors begin the discussion by using the term cross-national consistency. This can lead to confusion, as it can be confused with the existence of cross-national invariance. I suggest using the phrase: "the results are consistent with previous evidence obtained in other populations".
I agree with the Reviewer. Accordingly, in an entirely new Introduction (requested by Reviewer 2 who pointed out that it is “too brief and does not adequately address the interest of the study ...”), I have adopted the phrase suggested by Reviewer 1.
- I think it is important to mention to what extent the magnitude of the relationship between SWLS and general health is similar to previous studies. I also think it is necessary to use the term "perception of general health", which is not the same as "general health".
Again, I agree with the Reviewer. I also use the term perception of general health in the Introduction. I realized that this self-report is not a gold standard offered by an external individual. Accordingly, concurrent validity is not the correct term to use but construct validity.
- Finally, I believe that the use of a classical approach is not the only limitation of the study. It is necessary to incorporate some of the aspects mentioned above. Additionally, they mention the Rasch and/or Mokken analyses, but it is not clear why they do not present these results.
One of the co-authors (Dr Carrozzino) is a fervent/strong supporter of the Rasch/Mokken approach and “insisted” on having it mentioned as an alternative way of studying these issues. It was not that we carried out any such analyses, but we will in the future. We would have needed to split the samples to also carry out such analyses, which would have resulted in too small subgroups for the classical approach. Or, we would have needed an additional, new sample for doing such analyses.
Reviewer 2 Report
First of all, congratulations for the work carried out, it is a very interesting research and the methodology of the study is very well defined and developed. After reviewing the manuscript, it could be improved in the following points:
The introduction should be improved, it is too brief and does not adequately address the interest of the study and its rationale, moreover, few references are current (less than 5 years old).
In the "Materials and methods - Subjects and procedure" section, did participants sign an informed consent to participate in the study?
Among the references used in the manuscript, only 16 of the 42 are less than 5 years old. The number of current references should be increased.
Author Response
The introduction should be improved, it is too brief and does not adequately address the interest of the study and its rationale, moreover, few references are current (less than 5 years old).
Thank you for your response and opportunity to address the importance of the study with an extensive review of relevant studies and Introduction. Its length has been more than doubled and I am happy with the result.
In the "Materials and methods - Subjects and procedure" section, did participants sign an informed consent to participate in the study?
Yes, they obviously did provide their consent to participate in the stdy. This information has now been provided more thorougly (in yellow) in the same section.
Among the references used in the manuscript, only 16 of the 42 are less than 5 years old. The number of current references should be increased.
I have increased the number of References (less than 5 years old) from 16 to 30.
Reviewer 3 Report
Dear Authors, thank you for you research and contribution to this field.
Materials and Methods: "Ss with a Vietnamese nationality were invited to participate in a study" what does Ss means?
Table 1: Authors have described that the sample is composed with participants aging 14 to 64 years. In Table 1 there are only 2 groups for the dichotomized age variable, one for 14-24 yr and the other for 25-35 yr. What about the older participants, that are older than 35years? This frequency distribution must be corrected. This is only mentioned in Results.
Income: maybe the majority of readers are not aquaintenend with Vietnamese Dong (VND), therefore would be helpful to give the reader a comparison guide, maybe in US Dollars?
Statistical Analysis: which software was used to perform the CFA analysis? Please cite the software with versions, and if applicable the packages used.
Results: Figure 1 - Please add a legend for the arrows. What does the arrows from SWL to the items mean? And those only for the items? Figure 2 the same. COnsider also explain why some variables are in boxes and others in circles.
The acronym Ss is present in different sections in the text (for example again in 3.4) please explain what does that mean.
Author Response
Thank you for your review.
Materials and Methods: "Ss with a Vietnamese nationality were invited to participate in a study" what does Ss means?
Ss (subjects) means participants. We have changed Ss to participants, everywhere where Ss was used.
Table 1: Authors have described that the sample is composed with participants aging 14 to 64 years. In Table 1 there are only 2 groups for the dichotomized age variable, one for 14-24 yr and the other for 25-35 yr. What about the older participants, that are older than 35years? This frequency distribution must be corrected. This is only mentioned in Results.
In reply to Reviewer 1 who also raised this issue, one co-author noted: A better explanation of the age groups is added to the description of the cut-off points in the Measurement invariance of the Results section.
Income: maybe the majority of readers are not aquaintenend with Vietnamese Dong (VND), therefore would be helpful to give the reader a comparison guide, maybe in US Dollars?
Where the cut-off for income was introduced (Results section), we provided a comparison guide.
Statistical Analysis: which software was used to perform the CFA analysis? Please cite the software with versions, and if applicable the packages used.
As we comment in the article, CFA, measurement invariance, construct validity, and corrected item-total polyserial correlations were determined with Mplus 8.6 [19]. For determining descriptive statistics for the socio-demographic variables, for the SWLS items and total score, IBM SPSS 26 was used.
Results: Figure 1 - Please add a legend for the arrows. What does the arrows from SWL to the items mean? And those only for the items? Figure 2 the same. COnsider also explain why some variables are in boxes and others in circles.
Thank you very much for your suggestion. The variables that are represented in circles are the latent variables, and the observed variables (usually the items) are represented in squares. In this case, the graphical representation of a confirmatory factor analysis is done by drawing arrows that go from the latent variable to the items, indicating that those items are measuring that latent variable. Likewise, the observed variables (items) also contain a part of error that is estimated by the model. Each item contains its own error, represented by the arrow that goes towards each of them, and is represented on the right of the drawing. Usually this explanation is not offered in the article, it is enough to indicate the program used, the estimation method and the fit indices.
On the other hand, in the footnote of each figure it has been added that these are standardized coefficients.
The acronym Ss is present in different sections in the text (for example again in 3.4) please explain what does that mean.
As above, we have changed Ss to participants, everywhere where Ss was used.
Perhaps the answer prepared for another reviewer regarding the formed age groups can be added here. Or something similar...
When Irene and I read it, the first thing we thought was that this person knows what a confirmatory factor analysis is. Anyone who knows what the CFA is knows what the arrows mean... It has nothing to do with the figure in the article you send me. Actually they do the same as us in the model, although they include the drawing of the errors, which are also latent variables and are therefore represented with circles. Hardly anyone does. They offer it because the program they use offers it that way, with the drawing. Ours does not, but it is not necessary either. We have never been asked to add it.
I will try to explain it simply and in a way that you will not feel offended. Correct me if you consider it appropriate, please...
Reviewer 4 Report
1. Introduction
- Kindly review studies (if any) that examined the psychometric qualities of the (English version) SWLS in the Vietnamese context. It makes little sense to translate the scale if the original, English version has poor psychometric qualities.
- Kindly provide more information about the Vietnamese-language version of the SWLS (SWLS-V). Readers are curious whether any past studies examined the psychometric qualities of the SWLS-V.
- Some sentences are incomplete. Please check.
2. Materials and Methods
- What does Ss mean? If it refers to human participants, please use "participants" or "respondents".
- Kindly report the information about the ethical clearance of the research. As the sample consisted of individuals below 18 years old. Kindly report if parental/guardian consent is obtained.
- Kindly indicate the cut-off for the dichotomized groups (e.g., low vs. high income) in Table 1
- "For testing loading invariance, a change of ≥ -.010 in CFI", kindly take note that "a change of -.010" is different from "a change of .010". Please verify the suggested cut-off.
- Kindly explain the reason for measuring the effect size (Cohen's d).
3. Results
- Table 2, unbold the Mean and SD for items 1 and 2.
- Table 2, remove the underline for the row of low income.
- Figure 1, kindly indicates if the reported factor loadings are the standardized values.
- Sec. 3.4, as participants were sorted into two groups, it is not justifiable to exclude individuals aged 35 and above just because of the small percentage. The authors can use a different cut-off value for grouping.
- Kindly justify the cut-off value for income groups.
- Kindly justify why undergraduate bachelors are grouped under not in higher education.
- It is misleading to conclude that the model for all subgroups presented in Table 3 showed a good fit. The RMSEA value for some subgroups (e.g., Not in Higher Education) was greater than .80.
- kindly explain the meaning of the bolded values in Table 3.
- The authors reported that "The SWLS did not show metric invariance by age". It seems inaccurate because the three indicators are within the cut-off values. It should be scalar invariance that did not hold. Please check.
- Table 3, ΔCFI shall be interpreted as a change in CFI instead of CFI increase. The same goes for ΔRMSEA and ΔSRMR.
- "the CFI increased by 0.017". It should be decreased by 0.017.
- " a partial scalar model was tested keeping item 4 invariant" Do you mean to release the constraint of item 4?
4. Discussion
- The literature mentioned in the first paragraph shall be reviewed in the Introduction.
- Kindly discuss the findings of the reliability of the SWLS-V
- It is not clear how the fourth paragraph explains the differences in the measurement invariance result for education. Please elaborate on the point.
- Metric or scalar invariance in relation to age was downright poor?
- "and similarly for subjects in late adulthood" But those aged 36 and above were excluded from the study.
- "It is believed that the present findings may not have
been affected by the sizeable number of adolescent participants as previous studies [e.g., 38, 39] have supported the factorial and convergent validity of the SWLS with the relevant age group." This is misleading as the two cited studies were not conducted on Vietnamese samples. The generalizability is open. - The suggestion for a short form of SWLS requires further elaboration. As the short form was not tested in the present study, it is not clear whether a short form is invariant between subgroups.
The authors are suggested to present the conclusion in paragraph form.
Author Response
- Introduction
- Kindly review studies (if any) that examined the psychometric qualities of the (English version) SWLS in the Vietnamese context. It makes little sense to translate the scale if the original, English version has poor psychometric qualities.
- Kindly provide more information about the Vietnamese-language version of the SWLS (SWLS-V). Readers are curious whether any past studies examined the psychometric qualities of the SWLS-V.
In response to both a) and b), also Reviewer 1 made the same request as the Introduction was too brief and did not adequately address the interest of the study and its rationale. The request to give the manuscript more body with a review has now been met.
Previously, 4 studies have been published in which an SWLS-V was used, but they were not psychometric studies of the relevant SWLS-V. As indicated under “The Present Study”, only alpha’s were reported in those four studies. That’s it. Our study goes beyond only alpha (which, in addition, is also a sub-optimal measure of reliability).
- Some sentences are incomplete. Please check. Thank you; this check will be done.
- Materials and Methods
- What does Ss mean? If it refers to human participants, please use "participants" or "respondents".
Ss has been changed to participants everywhere in the text where it has been used.
- Kindly report the information about the ethical clearance of the research. As the sample consisted of individuals below 18 years old. Kindly report if parental/guardian consent is obtained.
This has been done in the Subjects and procedure section (in yellow).
- Kindly indicate the cut-off for the dichotomized groups (e.g., low vs. high income) in Table 1
Thanks for the suggestion. The cut-off points have been added in the Note of table 1
- "For testing loading invariance, a change of ≥ -.010 in CFI", kindly take note that "a change of -.010" is different from "a change of .010". Please verify the suggested cut-off.
Thanks for the suggestion. The negative sign was an error, it has been removed.
- Kindly explain the reason for measuring the effect size (Cohen's d).
An effect size makes cross-study comparisons possible, independent of sample size differences across studies.
- Results
- Table 2, unbold the Mean and SD for items 1 and 2.
It has been corrected
- Table 2, remove the underline for the row of low income.
Thank you, it has been another mistake. It has been removed.
- Figure 1, kindly indicates if the reported factor loadings are the standardized values.
Thank you again. We forgot to indicate it. Now this information has been added.
- Sec. 3.4, as participants were sorted into two groups, it is not justifiable to exclude individuals aged 35 and above just because of the small percentage. The authors can use a different cut-off value for grouping.
Thank you very much for your suggestion, but the problem in this sample is that more than 95% of the participants are 35 or younger, that is, the sample of this study is made up of very young people. We consider that it does not make much sense to include such a small percentage of people in a third group. Likewise, it would not be appropriate to include participants over 35 years of age in the second age group, since it is also a mostly young group. This group would be too biased.
- Kindly justify the cut-off value for income groups.
This has been justified in the Results section where the cut-off is introduced.
- Kindly justify why undergraduate bachelors are grouped under not in higher education.
Undergraduate graduates are university students; therefore, they have not completed their university studies. Perhaps it is more appropriate to describe the groups as “Bachelor degree or higher” and “No university degree”. It has been changed in the manuscript in the tables and the text.
- It is misleading to conclude that the model for all subgroups presented in Table 3 showed a good fit. The RMSEA value for some subgroups (e.g., Not in Higher Education) was greater than .80.
Although the RMSEA and CFI values are inconsistent in these cases, this can happen at times. These indices are commonly used to assess model fit, but CFI and RMSEA do not produce comparable qualitative assessments for any data set, as they are calculated differently. RMSEA is a non-standardized fit index, which makes it difficult to interpret, except if it is done using arbitrary cutoffs. But the CFI measures the relative improvement in fit (Shi et al., 2019). When RMSEA and CFI offer different assessments of the fit of the model, some authors argue that this does not mean that the model is poorly specified or that there is a problem with the data, but rather that these indices may differ in their interpretation because they assess the fit of the model from different perspectives (Lai and Green, 2016). Other authors indicate that, compared to the RMSEA, the SRMR, that is a standardized fit index, shows higher power to reject models that present poor fit to the data with ordinal responses (as in this case), regardless of the number of parameters to be estimated and the sample size (Shi et al., 2020). Therefore, the fit of the model can be evaluated, in this case, using the SRMR and the CFI. For these reasons, we consider that the one-factor model for the SWLS shows a good fit to the data in this sample.
Shi, D.; Lee, T.; Maydeu-Olivares, A. Understanding the Model Size Effect on SEM Fit Indices. Educ. Psychol. Meas. 2019, 79, 310–334, doi:10.1177/0013164418783530.
Lai, K.; Green, S.B. The Problem with Having Two Watches: Assessment of Fit When RMSEA and CFI Disagree. Multivar. Behav. Res. 2016, 51, 220–239, doi:10.1080/00273171.2015.1134306.
Shi, D.; Maydeu-Olivares, A.; Rosseel, Y. Assessing Fit in Ordinal Factor Analysis Models: SRMR vs. RMSEA. Struct. Equ. Modeling 2020, 27, 1–15, doi:10.1080/10705511.2019.1611434.
- kindly explain the meaning of the bolded values in Table 3.
Thanks for the suggestion. Added explanation in Table note.
- The authors reported that "The SWLS did not show metric invariance by age". It seems inaccurate because the three indicators are within the cut-off values. It should be scalar invariance that did not hold. Please check.
Thanks for the suggestion. Changed "metric" to "scalar"
- Table 3, ΔCFI shall be interpreted as a change in CFI instead of CFI increase. The same goes for ΔRMSEA and ΔSRMR.
Thanks for the suggestion. Changed "increase" to "change"
- "the CFI increased by 0.017". It should be decreased by 0.017.
Thanks for the suggestion. Changed "increased" to "decreased"
- " a partial scalar model was tested keeping item 4 invariant" Do you mean to release the constraint of item 4?
Thanks for the suggestion. The wording of all cases of partial scalar in the text has been revised.
- Discussion
- The literature mentioned in the first paragraph shall be reviewed in the Introduction
Indeed, this information has been moved to the Introduction section.
- Kindly discuss the findings of the reliability of the SWLS-V
This has been briefly mentioned now; and was indeed not referred to in the original manuscript.
- It is not clear how the fourth paragraph explains the differences in the measurement invariance result for education. Please elaborate on the point.
Thanks for the suggestion. A brief explanation has been added.
- Metric or scalar invariance in relation to age was downright poor ?
This should indeed be scalar invariance.
- "and similarly for subjects in late adulthood" But those aged 36 and above were excluded from the study.
Agree! This has been removed in the revision.
- "It is believed that the present findings may not have
been affected by the sizeable number of adolescent participants as previous studies [e.g., 38, 39] have supported the factorial and convergent validity of the SWLS with the relevant age group." This is misleading as the two cited studies were not conducted on Vietnamese samples. The generalizability is open.
Agree again. This mention has also been removed in the revision.
- The suggestion for a short form of SWLS requires further elaboration. As the short form was not tested in the present study, it is not clear whether a short form is invariant between subgroups.
On second thought, the argument that was offered, was weak. Also removed in the revision.
The authors are suggested to present the conclusion in paragraph form.
This has been carried out, with thanks to all your input.
As Irene indicates in another comment, we cannot do it because we do not know the income levels of the country, I seem to remember that you told us how to form the groups in some email...
We believe that it is not necessary to include all this explanation in the article since the objective in the models to study measurement invariance is not so much to see the fit of each model separately, as to see if there are differences among the nested models (configural, metric, scalar). We think that it can complicate the reading of the work, and we consider that it is better to leave it as a response to the reviewer. Perhaps a brief explanation can be added to the article, if you feel it is appropriate.
We think the reviewer questions this statement because there is really no age invariance. We think he wants it stated this way: The SWLS did not show scalar invariance by age.
And comment as you did...
Round 2
Reviewer 4 Report
I thank the authors for their efforts in revising the manuscript. There are two more comments for the authors:
1. After accepting the changes, check the manuscript and correct the spelling errors (e.g., Line 148, "the sprecified latent factor").
2. The authors changed the concurrent validity to construct validity in the title. However, in Lines 320 to 321, the authors wrote that "A structural equation model was specified to study the concurrent validity of the Vietnamese SWLS by using the General Health rating" (which makes sense to me. If the authors tested both concurrent and construct validity, it is not necessary to specify the type of validity in the title.
Author Response
Dear colleague,
Thanks again for providing excellent feedback to our work.
I have removed the spelling error to which you referred (changed to specified).
In addition, as we are dealing with a self-report measure of general health, not an external measure [gold standard], "concurrent" validity would not be suitable. So, where concurrent was still mentioned in the text, this was changed to 'construct'. This was also the reason why "concurrent" was changed to 'construct' in the title.
Thanks again.
Sincerely,
Willem A. Arrindell, PhD